# Transcriptome and Temporal Transcriptome Analyses in Single Cells

**DOI:** 10.3390/ijms252312845

**Published:** 2024-11-29

**Authors:** Jun Lyu, Chongyi Chen

**Affiliations:** Laboratory of Biochemistry and Molecular Biology, National Cancer Institute, National Institutes of Health, Bethesda, MD 20892, USA; jun.lyu@nih.gov

**Keywords:** single-cell transcriptome analysis, newly synthesized RNA detection, RNA metabolic labeling, single-cell temporal transcriptome analysis

## Abstract

Transcriptome analysis in single cells, enabled by single-cell RNA sequencing, has become a prevalent approach in biomedical research, ranging from investigations of gene regulation to the characterization of tissue organization. Over the past decade, advances in single-cell RNA sequencing technology, including its underlying chemistry, have significantly enhanced its performance, marking notable improvements in methodology. A recent development in the field, which integrates RNA metabolic labeling with single-cell RNA sequencing, has enabled the profiling of temporal transcriptomes in individual cells, offering new insights into dynamic biological processes involving RNA kinetics and cell fate determination. In this review, we explore the chemical principles and design improvements that have enhanced single-molecule capture efficiency, improved RNA quantification accuracy, and increased cellular throughput in single-cell transcriptome analysis. We also illustrate the concept of RNA metabolic labeling for detecting newly synthesized transcripts and summarize recent advancements that enable single-cell temporal transcriptome analysis. Additionally, we examine data analysis strategies for the precise quantification of newly synthesized transcripts and highlight key applications of transcriptome and temporal transcriptome analyses in single cells.

## 1. Introduction

Cells, as the basic functional units and building blocks of life, exhibit heterogeneity not only across cells of different types and functions, but also within seemingly identical cells. This cellular heterogeneity serves essential biological functions, such as contributing to drug resistance in bacteria and tumors or enabling the selection of color vision photoreceptors in Drosophila [1]. However, the functional diversity of individual cells is often obscured by population-averaged assays that are commonly used in biological studies. Traditional approaches, including microscopy, flow cytometry, and single-cell qPCR, have been employed to explore cellular heterogeneity, but they offer limited insight into specific cellular features [2].

In contrast, single-cell RNA sequencing (scRNA-seq), which profiles the transcriptome at the single-cell level, offers a comprehensive view of each cell’s state [3]. Over the past decade, significant improvements in transcript detection, accuracy, and cellular throughput have transformed scRNA-seq into the most powerful tool in dissecting cellular heterogeneity [4]. Its achievements range from identifying novel cell populations to mapping the human cell atlas [5].

Beyond mRNA profiling, efforts have been made to extend the scope of standard scRNA-seq in several directions. These advancements include the ability to profile the complete transcriptome that captures transcripts both with and without poly(A) tail [6,7,8,9]; the ability to map the spatial transcriptome that provides the spatial information of each transcript within tissue [10,11,12]; and the ability to profile the temporal transcriptome that offers the temporal information of each transcript, by distinguishing newly synthesized and nascent transcripts from pre-existing transcripts [13]. These innovations allow for a more comprehensive view at the single-cell level of the cellular state by probing not only the mRNAs, but also the noncoding RNAs and small RNAs, of the cell microenvironment by detecting the spatial transcriptome that reveals cell-to-cell interactions, and of the transcriptome kinetics by revealing the temporal transcriptome that reflects RNA transcription, processing, and decay.

Among these developments, temporal transcriptome profiling at the single-cell level (hereafter temporal scRNA-seq) has made remarkable progress in the field. It enables the detection of subtle transcriptomic changes and provides a framework for modeling RNA kinetics, with successful applications in immunology, neuroscience, cancer biology, and developmental biology [13]. Importantly, thanks to the newly synthesized transcriptome, temporal scRNA-seq offers a more accurate and robust dataset for cell fate trajectory inference compared to traditional scRNA-seq [13].

In this review, we summarize the methodological advancements in scRNA-seq, with a focus on the development, data analysis, and applications of temporal scRNA-seq. We aim to provide a solid foundation for readers new to scRNA-seq and temporal scRNA-seq, inspire innovation for developers, and discuss new application directions for advanced users exploring the potential of scRNA-seq and temporal scRNA-seq.

## 2. Development and Improvements of Single-Cell RNA Sequencing Methods

### 2.1. Methodologies for Constructing Amplifiable DNA Fragments from RNA (Figure 1A)

Single-cell RNA sequencing (scRNA-seq) comprises several critical steps, including single-cell isolation and lysis, RNA reverse transcription (RT), second-strand synthesis (SSS), cDNA amplification, library preparation, and sequencing. Due to the minute amount of RNA in a single cell, transcriptome amplification is necessary to generate sufficient material for library preparation. Therefore, the core steps of scRNA-seq are the steps of RT and SSS, which are required for most scRNA-seq protocols to convert RNA into amplifiable cDNA fragments. While RT is typically initiated by reverse transcriptase using a poly(T) primer that includes a stretch of specific sequence, SSS can be achieved through various approaches. In this section, we introduce four different SSS methods that facilitate the conversion of RNA into amplifiable cDNA, along with our LAST-seq method that uniquely and directly amplifies RNA prior to RT without performing SSS.

#### 2.1.1. SSS Enabled by Poly(A) Tailing

Poly(A) tailing is a sophisticated method which was first used for labeling the 3′ end of cDNA in the 1970s [14] and for cDNA cloning in the 1980s [15]. In 2009, Tang et al. applied this strategy in the first scRNA-seq experiment, utilizing terminal deoxynucleotidyl transferase (TdT) to add a poly(A) tail to the 3′ end of cDNA. This created a docking site for SSS using poly(T) primers containing a PCR anchor sequence [16]. In addition to poly(A) tailing, TdT can also catalyze the addition of other deoxynucleotides to the 3′ end of cDNA, such as poly(C) tailing [7]. However, the efficiency of the tailing process and the impact of different deoxynucleotide tailing on the efficiency of SSS remain unclear.

#### 2.1.2. SSS Mediated by Template-Switching

The template-switching reaction is a widely used method to add a PCR anchor sequence at the 3′ end of cDNA [17]. It leverages the TdT activity of M-MLV (Moloney Murine Leukemia Virus Reverse Transcriptase), which adds three consecutive cytosines (triple Cs) to the 3′ end of cDNA when the reverse transcriptase reaches the 5′ end of RNA. These triple Cs hybridize with a template-switching oligo (TSO), serving as a new template for the reverse transcriptase. Thanks to its DNA-dependent DNA polymerase activity, the enzyme further incorporates the reverse complement of the TSO sequence into the 3′ end of the cDNA, thereby creating a PCR anchor. This strategy was first demonstrated by Islam et al. in 2011 in STRT-seq and was later optimized by the Sandberg lab in the Smart-seq series [18,19,20,21,22]. Notably, aside from the triple Cs, the reverse transcriptase also incorporates other deoxynucleotides, accounting for approximately 50% of additional sequences [23,24]. As such, the reverse transcriptase may sometimes fail to complete template-switching, either by incorporating TSO-mismatched sequences or by prematurely dissociating due to complex secondary structures in the RNA, suggesting significant potential for future improvements in scRNA-seq methods based on template-switching-mediated SSS.

#### 2.1.3. SSS Mediated by RNase H and DNA Polymerase

The combination of RNase H and DNA polymerase was first used in the 1980s to synthesize the second strand of cDNA [25]. After reverse transcription, RNase H nicks the RNA strand in the RNA/DNA hybrid, and DNA polymerase extends the second-strand cDNA from the 3′ end of the RNA fragments, with DNA ligase filling any gaps. In 2012, Hashimshony et al. combined this SSS strategy with in vitro transcription (IVT) into a scRNA-seq assay called CEL-seq [26]. In this assay, a T7 promoter was attached to the 5′ end of first-stranded cDNA during RT, followed by SSS via RNaseH nicking and DNA polymerase extension, forming a double-stranded DNA template for IVT. Notably, the balance between nicking and extension is crucial for effective SSS, as excessive nicking can remove RNA fragments required for second-strand priming, while insufficient nicking also compromises DNA polymerase extension to complete SSS. The optimal conditions for the combination of RNase H and DNA polymerase to achieve efficient SSS remains unclear.

#### 2.1.4. SSS via Tn5 Tagmentation on RNA/cDNA Hybrids

The bacterial transposase Tn5 has been used to fragment double-stranded DNA (dsDNA) and add sequencing adaptors during library preparation [27]. The dimeric Tn5 transposon cuts dsDNA and adds the transposon DNA adaptors to the 5′ end of DNA fragments. In 2020, Di et al. discovered that Tn5 could also tagment DNA/RNA hybrids and add the transposon DNA adaptors to both DNA and RNA strands, leading to the development of the SHERRY assay [28]. However, in SHERRY, the DNA fragments are only amplifiable in the subsequent PCR step if two different adaptors are added to their ends, which results in a 50% loss of the original cDNA.

#### 2.1.5. Direct RNA Amplification Without RT/SSS

Converting RNA into amplifiable cDNA through RT and SSS is an essential yet often inefficient step in existing scRNA-seq methods, compromising overall RNA capture efficiency in single-cell transcriptome analyses. Recently, building on efficient IVT from single-stranded templates, we developed LAST-seq, a method that directly amplifies RNA prior to RT and SSS [29,30]. In LAST-seq, the RNA’s poly(A) tail is annealed to a LAST primer containing an rU-dT tail and a T7 promoter. The poly(A) annealing to rU anchors the RNA, while the short patch of poly(A) annealing to dT is nicked by RNase H and extended by DNA polymerase, thereby attaching the T7 promoter to the 3′ end of the RNA. This creates a single-stranded RNA template for IVT. With efficient T7 promoter tagging (~65%) and robust IVT linear amplification (hundreds-fold), we believe the LAST-seq approach could be highly advantageous in scenarios where RT and SSS are either infeasible or inefficient.

### 2.2. Improvements in RNA Capture Efficiency of Single-Cell RNA-Seq

RNA capture efficiency is fundamental to the performance of scRNA-seq methods. Low capture efficiency leads to inaccurate measurements and high levels of technical noise, compromising the ability to dissect cellular heterogeneity, such as cell type clustering and gene expression noise quantification (Figure 1B). In this section, we summarize optimizations and novel chemistry designs that have improved RNA capture efficiency in scRNA-seq methods.

The Smart-seq series has been improved by optimizing the chemistry at each step of the scRNA-seq workflow [19,20,21,22]. In Smart-seq2, cDNA yield was enhanced by (I) substituting a guanylate with a locked nucleic acid (LNA) guanylate at the 3′ end of TSO, (II) adding betaine and MgCl_2_ during RT, and (III) using KAPA HiFi Hot Start DNA Polymerase, which tolerates the RT buffer, for cDNA amplification. In Smart-seq3, further improvements were made in the RT step by (I) replacing the reverse transcriptase with Maxima H-minus reverse transcriptase, (II) switching the salt from KCl to NaCl or CsCl to reduce RNA secondary structures, (III) adding 5% PEG to promote molecular crowding, and (IV) adding GTPs or dCTPs to improve template-switching. As a result, the high-throughput version of Smart-seq3 achieved an RNA detection sensitivity of 68%, compared to 12.5% in a high-throughput Smart-seq2 derivative [31,32]. FLASH-seq, also derived from Smart-seq2, has claimed even higher detection sensitivity than Smart-seq3 by introducing key modifications, including (I) combining RT and cDNA preamplification in a single reaction, (II) using Superscript IV, and (III) increasing the dCTP concentration to enhance template-switching [33].

Likewise, the CEL-seq series has improved its RNA capture efficiency [34,35]. In CEL-seq2, improvements include (I) enhancing RT efficiency with more advanced reverse transcriptase, (II) switching from column-based to bead-based cDNA and RNA purification, and (III) incorporating a second PCR anchor into cDNA by random priming after IVT, thus eliminating inefficient RNA ligation. These optimizations resulted in an RNA capture efficiency of 19.7% or 22% in a miniaturized version, compared to 5.8% in the original CEL-seq [34]. CEL-seq+ further improved IVT efficiency through an optimized T7 promoter, achieving higher RNA detection sensitivity than CEL-seq2 [35].

Innovative chemistry designs have also significantly enhanced RNA capture efficiency in scRNA-seq. For example, in MATQ-seq, a multiple annealing strategy improved RT efficiency, with the specially designed MALBAC primer evenly annealing to RNA and priming cDNA synthesis at a low temperature. Poly(C) tailing was then performed on the 3′ end of the cDNA to drive efficient SSS. MATQ-seq reported a Poisson-distribution-corrected capture efficiency ranging from 76% to 102.4% [7]. For another example, LAST-seq improved capture efficiency by directly amplifying RNA prior to RT and SSS, achieving a capture efficiency of 35% [29]. Both methods, with their high capture efficiencies, are capable of detecting biological variations within homogenous cell populations.

### 2.3. Improvements in RNA Copy Number Quantification of Single-Cell RNA-Seq

The amplification of cDNA by polymerase chain reaction (PCR) is biased toward short and AT-rich sequences, leading to inaccurate RNA quantification in single cells. While IVT-mediated cDNA amplification mitigates PCR bias by using linear instead of exponential amplification, it does not allow for the direct quantification of RNA copy numbers. To address this, a unique molecular identifier (UMI) can be attached to cDNA during amplification, labeling individual RNA molecules with a random oligonucleotide sequence. This enables the identification and removal of duplicates arising from amplification, allowing for the direct and digital counting of RNA molecules [36,37] (Figure 1C).

The pioneering work in applying UMI to scRNA-seq was conducted by Islam et al. in STRT-seq, which successfully eliminated amplification bias [38]. Since then, UMI strategies have been widely adopted in most scRNA-seq methods [7,9,21,22,29,32,33,34,39,40,41,42,43,44,45,46,47]. Notably, UMI length plays a crucial role in quantification accuracy, in that shorter UMIs are prone to collision, leading to underestimation of the RNA copy number, while longer UMIs are more susceptible to sequencing errors, leading to overestimation (Figure 1C). A recent study using molecular spikes showed that only UMI lengths of 8 nt or longer provided accurate RNA counting across the full spectrum of expression levels, with a hamming distance of one set to tolerate sequencing errors for 8 nt UMIs [31].

While the UMI strategy removes amplification bias, it introduces a complication called strand invasion, which affects scRNA-seq methods that rely on template-switching to add UMIs at the 5′ end of RNA [48] (Figure 1D). Strand invasion occurs during RT when the UMI region of TSO hybridizes with the cDNA, affecting isoform detection and UMI counts. This issue was mitigated in FLASH-seq and Smart-seq3xpress by inserting a fixed spacer before the UMI region [22,33].

### 2.4. Improvements in Cellular Throughput of Single-Cell RNA-Seq (Figure 1E)

In addition to RNA capture efficiency and copy number quantification accuracy, the cellular throughput of scRNA-seq methods is crucial for many applications, such as dissecting tissue heterogeneity, profiling single-cell transcriptomes to create cell atlases, and performing CRISPR/Cas9-mediated single-cell perturbations [49]. Over the past decade, the scalability of scRNA-seq methods has increased from tens of cells to hundreds of thousands, revolutionizing biomedical research [50]. This section highlights advances in high-cellular-throughput scRNA-seq methods, discussing multiple approaches that elevate the scalability.

The earliest scRNA-seq methods profiled only tens of cells, manually pipetting individual blastomeres [16]. While effective for small-scale studies like early embryo development, scaling beyond this required new strategies. The Fluidigm C1 microfluidic system could capture up to 96 single cells and sequentially deposit their cDNA into individual microwells, achieving a cell capture efficiency of 39% [51,52]. Fluorescence-activated cell sorting (FACS) was used to distribute single cells into individual wells of PCR plates [18,20,21,22,26,28,29,33,34,53,54], enabling hundreds of cells to be profiled. In downstream scRNA-seq, individual cells could be barcoded by tagging the 3′ or 5′ end of cDNA before amplification or tagging the full length of cDNA after amplification, enabling the pooling of hundreds of single cells for sequencing.

To meet the growing demand for high-throughput scRNA-seq profiling, liquid-handling robots that automate PCR plate handling and reagent delivery have facilitated the profiling of thousands of cells [55]. More conveniently, combinatorial indexing methods, such as the sci-RNA-seq of Cao et al., uniquely label the transcriptomes of single cells or nuclei [41]. Specifically, fixed cells or nuclei are barcoded in two rounds of split-pooling, first during RT in 96 or 384 well plates containing barcoded poly(T) primers, and second during library preparation. This method allows the profiling of tens of thousands of cells in a single experiment, albeit with a cell recovery rate of less than 5%. Techniques like SPLiT-seq and sci-RNA-seq3 further increase cellular throughput by incorporating additional rounds of split-pool barcoding, pushing cellular throughput to two million cells [45,46].

On the other hand, FACS-based single-cell distribution has been replaced by microwell or microfluidic systems. In the microwell system, magnetic barcoded microbeads containing unique barcodes co-localize with cells for cell barcoding on the scale of thousands to tens of thousands of cells. This approach, first introduced in Cyto-seq and optimized in Seq-Well with a semipermeable membrane covering microwells to enable efficient buffer exchange, is further simplified by agarose-gel-based microwell preparation in Microwell-seq [39,42,44]. In the microfluidic system, cells, barcoded microbeads, and reaction buffers converge in one flow which is subsequently separated by oil, forming nanoliter droplet emulsions containing bead–cell pairs. Drop-seq and inDrop-seq, which evolved from Smart-seq and CEL-seq, respectively, are pioneering methods in this direction [32,40]. However, they can only profile thousands of cells within one hour due to random cell–bead convergence, with a cell capture efficiency of 2% and 26% in Drop-seq and inDrop-seq, respectively [56]. The commercial 10× Genomics Chromium platform multiplexes cell–bead encapsulation, enabling the profiling of tens of thousands of cells in minutes, with a cell capture efficiency between 50% and 80% [43].

Further innovations that combine combinatorial indexing with microfluidic systems, as demonstrated by scifi-RNA-seq and FIPRESCI that utilize Chromium ATAC-seq and Chromium RNA-seq reagents, respectively [57,58], further elevate the throughput to hundreds of thousands of cells. Specifically, cells are barcoded in 96 or 384 well plates before being encapsulated in the droplets for a second round of barcoding, achieving a pair of barcodes for individual cells.

Despite these advances to enhance the scalability of scRNA-seq, the required microfluidic devices are often inaccessible to many laboratories, motivating efforts to develop microfluidic-free approaches to generate nanoliter droplets. In a recent study, Clark et al. developed PIP-seq based on rapid templated emulsification of cells and barcoded hydrogel beads without the use of microfluidics [47]. A scalable number of nanoliter droplets encapsulating single cells, beads, and reagents could be generated by a two-minute vortex, providing a scalable and convenient method for scRNA-seq (Table 1).

## 3. Detection of Newly Synthesized RNA Using Single-Cell RNA Sequencing Method

### 3.1. Methodologies to Distinguish Between Newly Synthesized and Old Transcripts

Transcription is a dynamic process that can be dissected at the transcriptomic level by distinguishing newly synthesized transcripts from pre-existing ones. This is achieved by metabolic incorporation of noncanonical nucleotides into newly synthesized RNA after feeding cells with nucleoside analogs. The commonly used uridine analogs, such as 5-bromouridine (BrU), 5-ethynyluridine (5-EU), and 4-thiouridine (4sU) [59], each offer unique properties for RNA labeling and further biochemical enrichment or chemical conversion for mutation detection (Figure 2A).

BrU-based labeling. In this method, an anti-BrU antibody is used to enrich newly synthesized RNA containing BrU from pre-existing RNA [60,61,62]. BrU’s low cytotoxicity makes it suitable for long-term labeling, which is useful for RNA half-life measurement [63].5-EU-based labeling. In this method, 5-EU-labeled newly synthesized RNA is biotinylated via click chemistry, followed by enrichment with streptavidin beads [64,65]. While click chemistry is efficient and orthogonal to other cellular functional groups, the irreversible binding of streptavidin to biotin can compromise reverse transcription, limiting downstream applications [13].4sU-based labeling. The most commonly used analog, 4sU, has the advantage of reversible enrichment. After 4sU is incorporated into newly synthesized RNA, it can be biotinylated using either HPDP-biotin (N-[6-(Biotinamido)hexyl]-3′-(2′-pyridyldithio) propionamide-activated biotin) or MTS-biotin (methylthiosulfonate-activated biotin) via their activated disulfide groups [60,66,67,68,69]. HPDP-biotin offers a high specificity, enabling a low background from unlabeled pre-existing RNA. MTS-biotin offers a high sensitivity, increasing RNA yield by three-fold at the cost of a ten-fold higher background from unlabeled RNA compared to HPDP-biotin [70], thus it is suitable for detecting unstable RNA with a low abundance.

Overall, the biochemical enrichment strategy holds the advantage of sufficient sequencing of newly synthesized transcripts and compatibility with existing RNA-seq bioinformatic workflows. However, it requires a large input of total RNA, as newly synthesized RNA often represents only a small fraction of the total RNA pool. Additionally, multiple rounds of enrichment and washing cycles introduce technical variability between samples, requiring spike-ins for accurate data normalization.

In contrast to the biochemical enrichment strategy, the nucleotide chemical conversion strategy is enrichment-free, eliminating the need for a large input of total RNA. This enables the separation of newly synthesized RNA from pre-existing RNA in small input amounts, including single-cell RNA. The chemical conversion method identifies newly synthesized RNA in sequencing reads by detecting mutations introduced during reverse transcription at sites where nucleotide analogs have been incorporated and converted.

To date, several chemical treatments have been developed to convert 4sU into uridine analogs, cytidine analogs, or native cytidine, resulting in thymidine-to-cytosine mutations at 4sU sites in sequencing reads. For example, in SLAM-seq, iodoacetamide alkylates 4sU into an S-alkylated uridine derivative that is read as cytosine by reverse transcriptase [71]. In TUC-seq, osmium tetroxide and ammonia convert 4sU into native cytosine [72]. TimeLapse-seq uses sodium (meta)periodate and trifluoroethylamine to transform 4sU into a N4-trifluoroethyl-modified cytidine [73], while AMUC-seq employs acrylonitrile to generate an S-cyanoethylated 4-thiouridine derivative from 4sU [74]. In AENT-seq, hydrazine hydrate transforms 4sU into 4-hydrazino cytidine [75], and nRibo-seq uses a combination of sodium(meta) periodate and ammonium chloride to convert 4sU into native cytidine [76].

All of these methods demonstrate high and comparable conversion efficiency, with no evidence of compromising reverse transcription on oligonucleotide templates. However, a thorough and systematic comparison of their performance in practical samples is still lacking. In addition to 4sU conversion, TimeLapse-seq and TUC-seq have been extended to recode 6-thioguanosine into 2-aminoadenosine analogs and 6-hydrazino-2-aminopurine, respectively, leading to G-to-A mutations at 6sG sites in sequencing reads [77,78]. This enables applications where 4sU labeling is not suitable, such as studying the turnover of pseudouridylated RNAs or uridine-poor RNAs [79]. Moreover, dual labeling of 4sU and 6sG, combined with simultaneous conversion, allows for the precise measurement of RNA half-life and quantification of newly synthesized transcripts [78].

### 3.2. Methods for Temporal scRNA-Seq (Figure 2B)

Temporal scRNA-seq can be achieved by combining RNA metabolic labeling methods with scRNA-seq, allowing for the quantification of newly synthesized and pre-existing transcripts in single cells. For example, by coupling SLAM-seq with Smart-seq2, scSLAM-seq and NASC-seq were developed. scSLAM-seq reduces the chemical conversion time to 5 min, while NASC-seq shortens the 4sU labeling time to 15 min, while maintaining a high median signal-to-noise ratio [80,81]. However, since both methods require purifying total RNA from single cells in wells before reverse transcription and library preparation, some loss of the input RNA is inevitable. Indeed, the gene detection sensitivity was significantly improved in NACS-seq2, which bypasses this purification step [82].

SLAM-seq has also been integrated with high-cellular-throughput scRNA-seq methods, leading to several derivatives. These include SLAM-Drop-seq (based on Drop-seq), where chemical conversion is carried out in methanol-fixed cells with overnight incubation in methanol before microfluidic system loading, or under a mild condition (lower temperature and DMSO concentration) after mRNA is enriched by poly(T)-tagging resin beads [83,84]; sci-fate (based on sci-RNA-seq), where chemical conversion is carried out in formaldehyde-fixed cells before pool-split cycles [85]; and WELL-TEMP-seq (based on a microwell platform), where chemical conversion takes place in wells after cell–bead pairing in wells [86]. Recently, SLAM-seq was applied to methanol-fixed yeast cells, followed by transcriptome profiling using a 10× Genomics gene expression kit, demonstrating compatibility with this commercial platform [87]. Similarly, we developed NOTE-seq on a 10× Genomics platform. NOTE-seq employs SLAM-seq in formaldehyde-fixed human cells, followed by a modified 10× Genomics workflow [88].

Alternatively, TimeLapse-seq has been combined with Drop-seq, where U-to-C conversion is performed after mRNA enrichment on poly(T)-tagging resin beads, resulting in scNT-seq [89]. scNT-seq demonstrated that TimeLapse-seq outperformed SLAM-seq in a one-pot reaction on pooled beads, though a clear explanation remains elusive. One possibility is that the high DMSO concentration in SLAM-seq significantly lowers the melting temperature between the poly(A) tail and the poly(T) oligo, causing mRNA to detach from the beads and be lost during washing.

In addition, scEU-seq (based on CEL-seq2) and LET-seq (based on scEU-seq) leverage 5-EU labeling of newly synthesized RNA for in situ biotinylation via click chemistry, followed by cellular barcoding during reverse transcription, allowing the physical separation of new transcripts from the pre-existing ones before sequencing [65,90]. scEU-seq requires spike-ins for each cell and can detect significantly high UMI counts after very short labeling times (15–30 min).

While a number of temporal scRNA-seq methods have been developed, a systematic comparison between them is still lacking. Nevertheless, temporal scRNA-seq enhances the capability of traditional scRNA-seq and offers significant advantages in data analysis.

### 3.3. Data Analysis of Temporal scRNA-Seq (Figure 2C)

The primary steps of temporal scRNA-seq data analysis are largely the same as those of standard scRNA-seq. These include converting image data into sequence data (fastq data calling), aligning reads to the reference genome and annotated transcripts (read mapping), assigning reads to individual cells (cell demultiplexing), and quantifying RNA molecules (UMI counting) [91]. This standard pipeline is adopted by methods that physically separate newly synthesized and pre-existing transcripts.

However, specialized computational steps are required for methods relying on chemical conversion to distinguish newly synthesized and pre-existing RNA. Specifically, UMI-represented reads containing T-to-C mismatches are classified as newly synthesized transcripts, while the remaining reads are assigned to pre-existing transcripts [85,89]. Notably, T-to-C mismatches can also arise from single-nucleotide polymorphisms (SNPs) between the experimental cells and the reference genome. To address this, SNP background masking is applied to exclude overlapping T-to-C mismatch sites. The SNP profile of the experimental cells can be obtained from public databases or from 4sU-negative, treatment-free cells sequenced in parallel.

While SNPs can inflate the number of T-to-C mismatch sites, artificially increasing the counts of newly synthesized transcripts, the limited incorporation rate of 4sU, typically ranging from 1% to 10%, leads to a lack of T-to-C mismatches in a significant portion of newly synthesized transcripts [59]. This underestimation is especially evident for transcripts with fewer uridines than average. One approach to address this issue (referred to as the “factor scaling strategy”) is to scale the newly synthesized transcript counts by cell-specific scaling factors, also known as detection rates. These detection rates can be estimated from a group of genes where the difference in total RNA levels equals the difference in newly synthesized RNA levels after a brief period of stimulation. As a result, the ratio of the difference in newly synthesized RNA to total RNA reflects the detection rate [85]. However, as this is a global factor, gene-specific biases are inevitable.

A more accurate approach (referred to as the “statistical correction strategy”) is to model the number of T–C mismatches per count with a binomial mixture distribution. This allows for the estimation of the fraction of newly synthesized transcripts among all RNAs for individual genes by maximum likelihood or Bayesian methods [73,80,81,92]. As a result, new transcript counts can be computed from total RNA counts for each gene. In reality, due to the limited sequencing coverage in single-cell transcriptome sequencing, applying this model to every gene across tens of thousands of cells can be experimentally and computationally challenging.

To overcome this limitation, a third approach (referred to as the “mixed strategy”) combines elements of both the statistical correction strategy and the detection rate concept from the factor scaling strategy [86,88,89]. In this approach, new transcript counts for each gene across cells of the same population are calculated using the statistical correction strategy. These counts are then aggregated and compared to the observed new transcript counts to calculate the detection rates for each cell. The corrected newly synthesized transcript counts for every gene in the individual cells are subsequently determined based on the detection rate and compared to the respective total RNA counts to establish the final expression levels of newly synthesized transcripts. Pipelines implementing these strategies include sci-fate [85] for factor scaling, NASC-seq [81] and scSLAM-seq [80] for statistical correction, and scNT-seq [89], WELL-TEMP-seq [86], and NOTE-seq [88] for the mixed strategy (Table 2).

### 3.4. Applications of Temporal scRNA-Seq

Temporal scRNA-seq outperforms traditional scRNA-seq by providing high temporal resolution for detecting rapid transcriptional changes occurring shortly after stimulation. It also enables the dissection of RNA kinetics, including transcription, RNA processing, and RNA decay, thereby offering significant potential for addressing fundamental biological questions ranging from gene expression regulation to embryonic development (Figure 2D). In addition, temporal scRNA-seq enhances single-cell data analysis, including cell classification and trajectory inference of cell fate (Figure 2E). In this section, we review various applications from temporal scRNA-seq studies conducted to date.

One obvious advantage of measuring newly synthesized transcripts is the ability to monitor rapid changes in the transcriptome that are often undetectable at the total transcript level. For example, NASC-seq and NOTE-seq were used to study the rapid process of early-stage T-cell activation, revealing differentially expressed mitochondrial genes and transcription factors at the level of newly synthesized transcripts, respectively [81,88]. scSLAM-seq was employed to investigate the onset of infection with lytic cytomegalovirus in single mouse fibroblasts, describing cell responses through newly synthesized transcripts [80]. WELL-TEMP-seq exploited the newly synthesized transcripts to depict the early-stage gene-expression dynamics of colorectal cancer cells treated with an anti-tumor drug [86]. In addition, measuring new transcripts improves the classification of continuous cell states. In this context, sci-fate and NOTE-seq demonstrated that a combined analysis of newly synthesized and total transcriptomes allows for the classification of activated cells across several time points into distinct subpopulations that are otherwise masked at the total transcriptome level [85,88]. Furthermore, scSLAM-seq showed that the ratio of newly synthesized transcripts to total transcripts could effectively separate uninfected from infected cells with high precision [80].

Another advantage of temporal scRNA-seq is to infer RNA synthesis and decay rates. For example, scEU-seq employs pulse-chase experiments alongside RNA dynamic models to accurately estimate RNA synthesis and degradation rates, revealing distinct RNA regulatory strategies during the cell cycle and the differentiation of intestinal stem cells [65]. Similarly, scNT-seq determined RNA biogenesis and decay rates to uncover RNA regulatory strategies during the stepwise conversion between pluripotent and rare totipotent two-cell embryo (2C)-like stem cell states [89]. In a study on zebrafish embryogenesis, zygotic newly synthesized transcripts were labeled, enabling the distinction from maternal transcription and facilitating the study of the fate of individual maternal transcripts [93]. Moreover, through pulse experiments with varying durations and RNA kinetic modeling, RNA transcription and degradation rates were quantified within individual cell types during their specification, revealing maternal and zygotic mRNA regulatory kinetics [84].

Additionally, newly synthesized transcripts have recently been used to infer bursting kinetics, demonstrating that analysis of new transcripts can separate kinetic parameters that specify burst size, showing that the synthesis rate controls burst size. Furthermore, these transcripts have been used to infer transcriptome-wide co-bursting at the allelic resolution, indicating that co-bursting rarely occurs more frequently than expected by chance regardless of genomic distance, except for in certain gene pairs [82] (Table 3).

On the other hand, single-cell analysis, especially the trajectory inference of cell fate, can be significantly improved by temporal scRNA-seq. Traditional scRNA-seq data typically contain 15–25% intronic reads, indicating unspliced transcripts. This information can be utilized to construct RNA velocity (the time derivative of RNA expression), which predicts the future states of cells on a timescale of hours [94,95]. While RNA velocity has proven beneficial in many studies, its accuracy is limited by the randomness of capturing intronic reads, which results from varying splicing rates and intron stability. Replacing intronic reads with newly synthesized transcripts can overcome this limitation, providing RNA velocity with a clear directionality and a controllable temporal resolution. A computational approach named Dynamo has been developed to construct RNA velocity using metabolic labeling information, demonstrating its superiority in many applications [86,89,96]. Alternatively, traditional scRNA-seq data can be used to infer the order of biological progression by learning a graph manifold of single cells based on transcriptome similarity [97]. This trajectory reflects the central trend of biological progression within cell populations rather than individual cells. In contrast, newly synthesized transcripts can bridge cell-to-cell trajectories between individual cells sampled at different time points, utilizing the cell states inferred from the total transcriptome [85] (Table 4).

### 3.5. Technical Limitations and Solutions

Although the scRNA-seq has laid a solid technical foundation for temporal scRNA-seq, incorporating 4sU labeling and downstream procedures compromises its technical performance. For instance, in methods practicing long-time in situ chemical conversion in methanol, RNA leakage may occur, leading to RNA loss and barcode swap among cells. This caveat could be avoided by conducting in situ chemical conversion in formaldehyde-fixed cells where RNA is cross-linked and confined in cells [85,88]. In methods conducting template-switching (T-S), compromised progress of reverse transcriptase due to 4sU incorporation and chemical conversion fails T-S, lowering the overall RNA capture efficiency. This caveat could be rescued by a second run of RT using random priming primers containing 5′ end TSO, or avoided at the beginning by replacing T-S with direct RNA amplification [29,89].

Additionally, the 4sU incorporation rate significantly influences the ultimate data quality, with a high rate compromising read mapping and a low rate leading to a sparse data matrix of newly synthesized RNA. An optimal incorporation rate ranging from 1% to 10% could result from titrations of 4sU with a concentration of 100 µM to 1 mM and a labeling time of 15 min to 24 h, with the consideration of cytotoxicity and RNA half-life, respectively [59].

Finally, increasing the sequencing length is always helpful for detecting newly synthesized RNA from a bioinformatics perspective [59].

## 4. Concluding Remarks and Perspectives

Single-cell RNA sequencing (scRNA-seq) methods have evolved significantly over the past decade, with innovations enhancing RNA capture efficiency, RNA copy number quantification accuracy, and cellular throughput. Recent advancements have incorporated temporal information into scRNA-seq through RNA metabolic labeling, along with physical or computational separation to distinguish between newly synthesized and pre-existing RNA. Temporal scRNA-seq provides new insights into single-cell transcriptomics, particularly in studies of rapid cellular processes, embryogenesis, and genome-wide RNA kinetics, where temporal information reflecting real-time transcriptional activity is crucial.

While most single-cell temporal transcriptome studies have been conducted in vitro or ex vivo, applying temporal scRNA-seq to in vivo systems is also feasible. For example, in uracil phosphoribosyltransferase (UPRT) transgenic mice, UPRT is expressed in a cell-type-specific manner, enabling the conversion of 4-thiouracil into 4sU monophosphate and its subsequent incorporation into newly synthesized transcripts [98,99]. This approach allows the investigation of physiological and pathological processes in mice using temporal scRNA-seq. Furthermore, incorporating spatial information alongside temporal scRNA-seq could help unravel spatial-temporal transcriptomes at the tissue level, elucidating potential connections between cellular environments and the transcriptomic responses of single cells [100].

At the cellular level, a fundamental process to investigate is gene expression, which is regulated by many factors, including transcriptome factors, 3D genome organization, DNA supercoiling, chromatin accessibility, and histone and DNA modifications. Since newly synthesized transcripts reflect real-time transcriptional changes, temporal scRNA-seq can be integrated with other single-cell omics assays to study the interplay between transcriptional activity and various chromatin features.

Recently, single-cell CRISPR screening has been integrated with temporal scRNA-seq to identify key regulators of transcriptome kinetics [101]. This combination is anticipated to lead to broader applications in functional genomics research. Additionally, third-generation sequencing technologies enable full-length sequencing and direct RNA sequencing [102], offering great potential to further enhance detection sensitivity and streamline workflows for transcriptome and temporal transcriptome analyses in single cells.

## Figures and Tables

**Figure 1 ijms-25-12845-f001:**
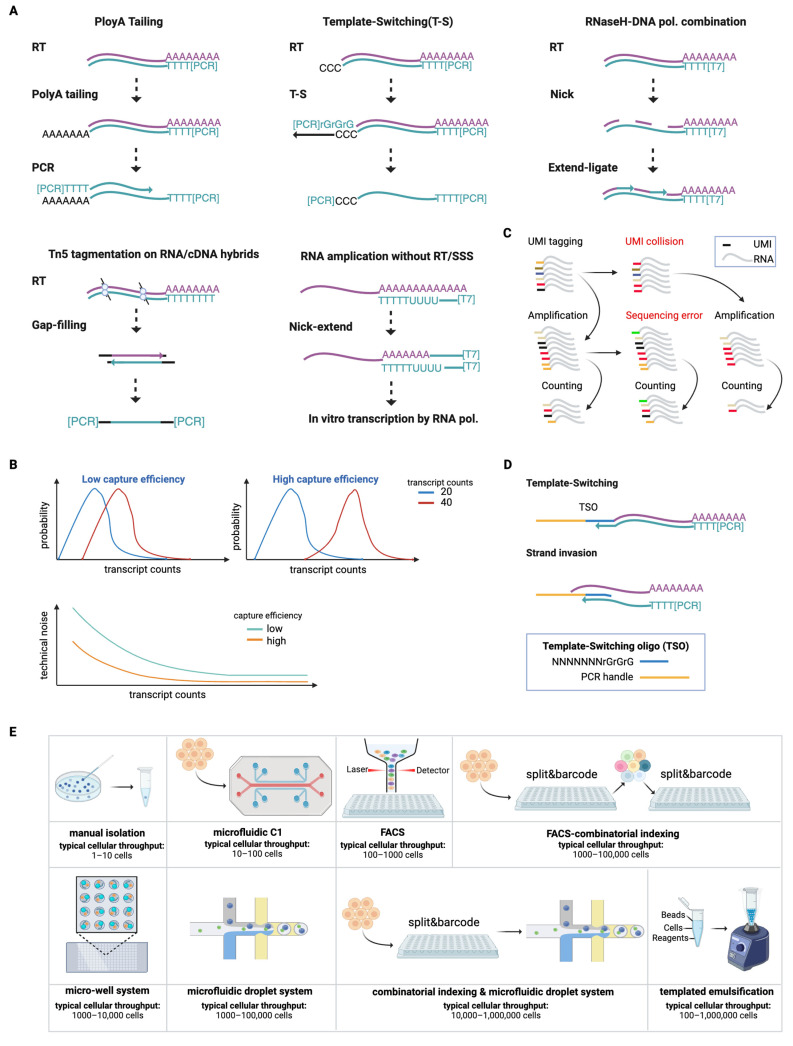
Methodologies of scRNA-seq methods. (**A**) Methodologies for constructing amplifiable DNA fragments from RNA. (**B**) Impacts of capture efficiency on RNA quantification and technical noise. (**C**) RNA quantification by UMI and UMI miscounting due to UMI collision and sequencing errors. (**D**) Strand invasion during template-switching. (**E**) Methodologies to increase cellular throughput of scRNA-seq.

**Figure 2 ijms-25-12845-f002:**
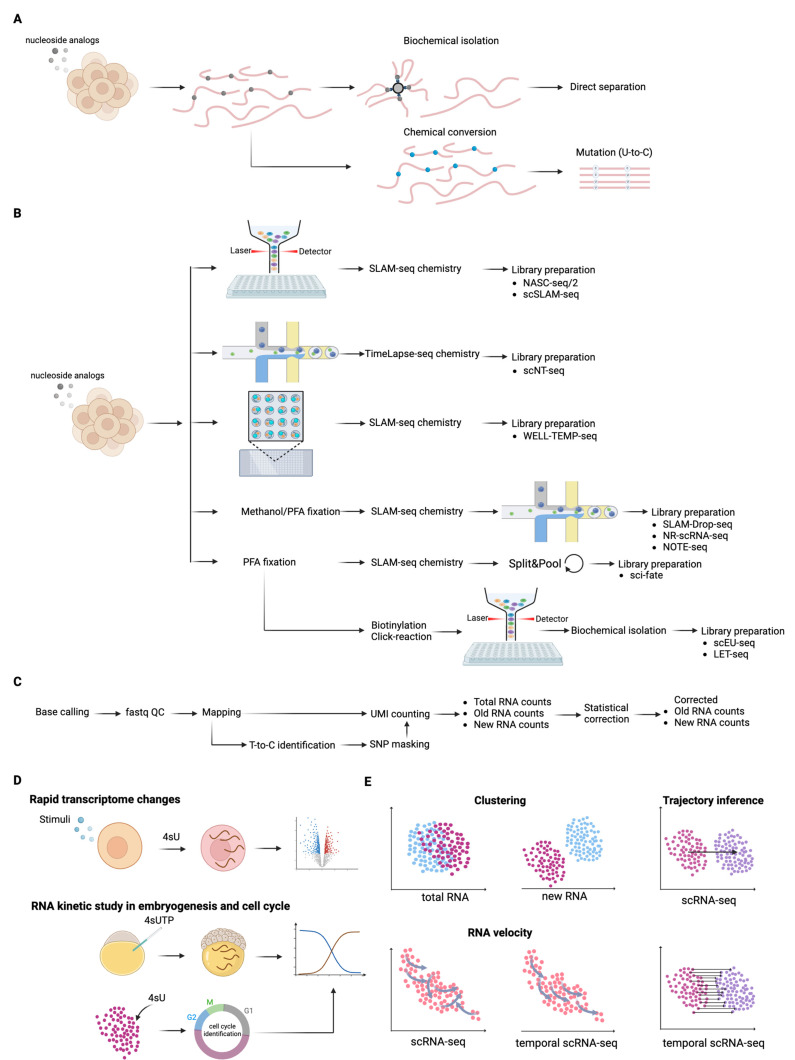
Methodologies and applications of temporal scRNA-seq methods. (**A**) Methods to separate newly synthesized transcripts and pre-existing transcripts. (**B**) Combination of RNA metabolic labeling and scRNA-seq. (**C**) Data analysis pipeline of temporal scRNA-seq. (**D**) Temporal single-cell transcriptome analysis captures acute gene expression changes in individual cells upon rapid stimulation and models RNA kinetics during embryogenesis and between cell cycle stages. (**E**) Temporal scRNA-seq improves the resolutions of cell clustering and trajectory inference, and the directionality of RNA velocity inference.

**Table 1 ijms-25-12845-t001:** Summary of single-cell RNA-seq methods introduced in this review.

Single-Cell RNA-Seq Methods Introduced in This Review *							
Methods	SSS Strategy	Amplification Strategy	Counting	Technical Improvements	Limitations	Debut	References
single-cell mRNA-seq	polyA tailing	PCR	full-length	first method demonstrating RNA-seq at single-cell level	low throughput, PCR amplification bias	2009	[16]
STRT-seq	template-switching	PCR	5′ end	cell multiplexing with cellular barcode	low throughput, PCR amplification bias	2011	[18]
Smart-seq/2	template-switching	PCR	full-length	single-cell full-length sequencing of mRNA	low throughput, PCR amplification bias	2012/2013	[19,20]
CEL-seq	RNase H and DNA polymerase	IVT	3′ end	reduced amplification bias via linear amplification of cDNA	low throughput, low RNA capture efficiency	2012	[26]
STRT-seq (UMI)	template-switching	PCR	5′ end	accurate quantification via UMI counting	low throughput	2014	[38]
MARS-seq	RNase H and DNA polymerase	IVT	3′ end	enhanced cellular throughput	limited RNA capture efficiency	2014	[55]
Drop-seq	template-switching	PCR	3′ end	first method incorporating microfluidic droplet system with scRNA-seq	low cell–bead co-capsulation efficiency	2015	[32]
inDrop-seq	RNase H and DNA polymerase	IVT	3′ end	co-first method incorporating microfluidic droplet system with scRNA-seq	limited cell–bead co-capsulation efficiency	2015	[40]
Cyto-seq	template-switching	PCR	3′ end	first method incorporating microwell system with scRNA-seq	potential cell size bias	2015	[39]
CEL-seq2	RNase H and DNA polymerase	IVT	3′ end	enhanced RNA capture efficiency	low throughput	2016	[34]
MATQ-seq	polyC tailing	PCR	full-length	enhanced RNA capture efficiency	low throughput	2017	[7]
10× Genomics Chromium platform	template-switching	PCR	3′ end and 5′ end	commercially available, enhanced cellular throughput	costly	2017	[43]
Seq-Well	template-switching	PCR	3′ end	enhanced reaction efficiency via buffer exchange	potential cell size bias	2017	[42]
sci-RNA-seq, SPLiT-seq	template-switching	PCR	3′ end	high cellular throughput based on FACS and split-pool	cell loss during split-pool	2017, 2018	[41,45]
Microwell-seq	template-switching	PCR	3′ end	streamlined microwell preparation	potential cell size bias	2018	[44]
sci-RNA-seq3	template-switching	N.A	3′ end	ultra-high cellular throughput with FACS being replaced by dilution	cell loss during split-pool	2019	[46]
Smart-seq3	template-switching	PCR	full-length	full-length sequencing with UMI counting, enhanced RNA capture efficiency	low throughput	2020	[21]
SHERRY	Tn5 tagmentation on RNA/cDNA hybrids	PCR	full-length	rapid library-prep protocol	limited RNA capture efficiency	2020	[28]
scifi-RNA-seq	template-switching	PCR	3′ end	ultra-high cellular throughput, 10× Genomics Chromium platform compatible	costly	2021	[57]
Smart-seq3xpress	template-switching	PCR	full-length, 5′ end	enhanced RNA capture efficiency, strand invasion prevention	medium throughput	2022	[22]
FLASH-seq	template-switching	PCR	full-length, 5′ end	enhanced RNA capture efficiency, strand invasion prevention, fast library-prep protocol	medium throughput	2022	[33]
LAST-seq	direct RNA amplification	IVT	3′ end	enhanced RNA capture efficiency, low technical noise	customized primer, low throughput	2023	[29]
FIPRESCI	template-switching	PCR	5′ end	ultra-high cellular throughput, 10× Genomics Chromium platform compatible	costly	2023	[58]
PIP-seq	template-switching	PCR	3′ end	fast and microfluidic-free droplets generation, throughput highly scalable	limited RNA capture efficiency	2023	[47]

* To concentrate on our topic, some excellent methods are not included in this review.

**Table 2 ijms-25-12845-t002:** Temporal single-cell RNA-seq data analysis pipeline in this review.

Temporal Single-Cell RNA-Seq Data Analysis Pipeline Mentioned in This Review *			
Data Analysis Pipeline	Strategy	GitHub Link	Reference
sci-fate pipeline	factor scaling	https://github.com/JunyueC/sci-fate_analysis (accessed on 25 November 2024)	[85]
NASC-seq pipeline	statistical correction	https://github.com/sandberg-lab/NASC-seq (accessed on 25 November 2024)	[81]
scSLAM-seq pipeline	statistical correction	http://software.erhard-lab.de (accessed on 25 November 2024)	[80]
scNT-seq pipeline	mixed strategy	https://github.com/wulabupenn/scNT-seq/ (accessed on 25 November 2024)	[89]
WELL-TEMP-seq pipeline	mixed strategy	https://github.com/songjiajia2018/Well-TEMP-Seq (accessed on 25 November 2024)	[86]
NOTE-seq pipeline	mixed strategy	https://github.com/lyuj2022/NOTE-seq (accessed on 25 November 2024)	[88]

* To stick with the content, representative data analysis pipelines are listed in this table.

**Table 3 ijms-25-12845-t003:** Summary of amplification of temporal scRNA-seq in this review.

Amplification of Temporal scRNA-Seq				
Methods	Technical Features	Applications	Debut	Reference
scSLAM-seq	chemical-conversion-coupled metabolic-labeling sequencing at single-cell level	investigates the onset of infection with lytic cytomegalovirus in single mouse fibroblasts	2019	[80]
NASC-seq	full-length sequencing of metabolic labeled mRNA	reveals differentially expressed mitochondrial genes during early stage of T cell activation	2019	[81]
scEU-seq	biochemical isolation of newly synthesized RNA at single-cell level	reveals distinct RNA regulatory strategies during the cell cycle and differentiation of intestinal stem cells	2020	[65]
sci-fate	formaldehyde fixation compatible, split-pool, high throughput	quantifies the dynamics of the cell cycle and glucocorticoid receptor activation	2020	[85]
scNT-seq	microfluidic droplet system, high throughput	uncovers RNA regulatory strategies during the stepwise conversion between pluripotent and rare totipotent two-cell embryo (2C)-like stem cell states	2020	[89]
N.A.	10× Genomics compatible, methanol fixation, high throughput	distinguishes zygotic newly synthesized transcripts and maternal transcripts during zebrafish embryogenesis	2021	[93]
WELL-TEMP-seq	microwell system compatible, high ratio of cell–bead pairing	depicts the early-stage gene expression dynamics and gene regulatory network of colorectal cancer cells treated with an anti-tumor drug	2023	[86]
SLAM-Drop-seq	microfluidic droplet system, chemical conversion in methanol, high throughput	depicts gene-specific kinetic rates during the cell cycle	2023	[83]
N.A.	microfluidic droplet system, chemical conversion in mild condition, high throughput	reveals maternal and zygotic mRNA regulatory kinetics through RNA kinetic modeling during zebrafish embryogenesis	2024	[84]
LET-seq	adapted from scEU-seq, capture both polyA+ and polyA− newly synthesized RNAs	reveals dynamic transcriptional reprogramming during zygotic genome activation in mice	2024	[90]
NR-scRNAseq	10× Genomics compatible, methanol fixation, high throughput	measures gene specific transcriptional noise and the fraction of active genes in *S. cerevisiae*.	2024	[87]
NOTE-seq	10× Genomics compatible, formaldehyde fixation, high throughput	reveals differentially expressed transcription factors during early stage of T cell activation	2024	[88]
NASC-seq2	advanced version of NASC-seq with enhanced RNA capture efficiency	infers bursting kinetics and transcriptome-wide co-bursting at the allelic resolution	2024	[82]

N.A., not applicable.

**Table 4 ijms-25-12845-t004:** Data analysis comparison between scRNA-seq and temporal scRNA-seq.

Data Analysis Comparison Between scRNA-Seq and Temporal scRNA-Seq		
Analysis	scRNA-Seq	Temporal scRNA-Seq
**Cell clustering based on**	transcriptome	transcriptome
		newly synthesized transcriptome
		ratio of newly synthesized transcriptome to transcriptome
		combined transcriptome (transcriptome joined on newly synthesized transcriptome)
**Trajectory analysis based on**		
I. RNA velocity	intronic transcripts	newly synthesized transcripts, intronic transcripts
II. Transcriptome similarity	transcriptome	past transcriptome, transcriptome
	population-to-population trajectory	cell-to-cell trajectory

## Data Availability

This study did not generate any unique datasets or codes.

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
