# Peer review of "Transcriptome and Temporal Transcriptome Analyses in Single Cells"

_ijms, 2024, doi:10.3390/ijms252312845_

Round 1

Reviewer 1 Report

Comments and Suggestions for Authors

In this manuscript, Lyu et al. provide an overview of single-cell RNA sequencing (scRNA-seq) techniques, with a particular focus on the integration of metabolic labeling, which is a highly relevant and exciting topic in the field. This approach is a valuable tool for studying RNA dynamics in heterogeneous biological systems, and the authors highlight the novel applications of this technique. Overall, the manuscript presents important advancements in the field.

However, my main concern lies with the overlap between this manuscript and the existing review by Erhard et al. (2022) published in Nature Reviews Methods Primers, which extensively covers metabolic labeling scRNA-seq approaches. It appears that certain sections, particularly those describing metabolic labeling in scRNA-seq, have significant similarities to the work by Erhard et al. It would be beneficial if the authors could further differentiate their review by emphasizing aspects that have not been previously discussed in the Erhard paper.

Additionally, the timeline of studies that have utilized metabolic labeling in scRNA-seq needs clarification. For example, the manuscript claims that NOTE-seq integrates in-situ IAA chemical treatment with 10x Genomics, yet the paper by Holler et al. (Nature Communications, 2021) had already implemented in-situ IAA treatment with 10x Genomics in zebrafish embryos. It would strengthen the manuscript if the authors could better contextualize these studies and clarify how their discussion contributes novel insights.

As a review paper, it would be highly beneficial to provide a detailed summary of the various scRNA-seq technologies, alongside a clear comparison of the advantages and limitations of each approach. This would offer readers a comprehensive understanding of the field and help them identify the key features of each method. Additionally, discussing potential areas for improvement, particularly in terms of enhancing the performance of metabolic labeling parameters, such as improving RNA recovery rates or optimizing other technical aspects, would add significant value to the review.

Author Response

In this manuscript, Lyu et al. provide an overview of single-cell RNA sequencing (scRNA-seq) techniques, with a particular focus on the integration of metabolic labeling, which is a highly relevant and exciting topic in the field. This approach is a valuable tool for studying RNA dynamics in heterogeneous biological systems, and the authors highlight the novel applications of this technique. Overall, the manuscript presents important advancements in the field.

We thank the reviewer’s comment that our manuscript presents important advancements in the field. We sincerely appreciate the reviewers’ constructive suggestions to further improve this manuscript. We provide a point-to-point response as below.

However, my main concern lies with the overlap between this manuscript and the existing review by Erhard et al. (2022) published in Nature Reviews Methods Primers, which extensively covers metabolic labeling scRNA-seq approaches. It appears that certain sections, particularly those describing metabolic labeling in scRNA-seq, have significant similarities to the work by Erhard et al. It would be beneficial if the authors could further differentiate their review by emphasizing aspects that have not been previously discussed in the Erhard paper.

We agree with the reviewer’s comment that the sections covering temporal scRNA-seq in this review have certain overlap with the review by Erhard et al. (2022) published in Nature Reviews Methods Primers, which is an excellent and extremely comprehensive review, making it difficult, if not impossible, for us to avoid any overlap between our review and the existing one while still comprehensively covering the current status and progress in the field of temporal scRNA-seq. Nevertheless, we updated the readers with recent progresses after 2022. In the revision, we have also added an additional section focusing on the technical limitations and solutions of temporal scRNA-seq, providing perspectives that differentiate from the topics in the Erhard review (Line 498-516).

Additionally, the timeline of studies that have utilized metabolic labeling in scRNA-seq needs clarification. For example, the manuscript claims that NOTE-seq integrates in-situ IAA chemical treatment with 10x Genomics, yet the paper by Holler et al. (Nature Communications, 2021) had already implemented in-situ IAA treatment with 10x Genomics in zebrafish embryos. It would strengthen the manuscript if the authors could better contextualize these studies and clarify how their discussion contributes novel insights.

We thank the reviewer for the suggestion of listing the timeline of studies, and clarifying the differences and similarities between NOTE-seq and the Holler paper. In the revision, we have added a Table 3 listing the timeline of published temporal scRNA-seq methods and their applications. Instead of focusing on how their discoveries contribute novel insights scientifically, in this review we focused on what these methods can offer technically.

As a review paper, it would be highly beneficial to provide a detailed summary of the various scRNA-seq technologies, alongside a clear comparison of the advantages and limitations of each approach. This would offer readers a comprehensive understanding of the field and help them identify the key features of each method.

We thank the reviewer for the suggestion of providing a detailed summary of various scRNA-seq methods. We have added a Table 1 listing technical improvements and limitations of existing scRNA-seq methods.

Additionally, discussing potential areas for improvement, particularly in terms of enhancing the performance of metabolic labeling parameters, such as improving RNA recovery rates or optimizing other technical aspects, would add significant value to the review.

We thank the reviewer for the suggestion of discussing potential areas for future technical improvement. The scRNA-seq methods have been extensively improved over the past more than a decade, without much room for further major improvements. In contrast, the methods of temporal scRNA-seq have just emerged over the past a few years and we can foresee significant improvements in the coming years. In the revision, we have added an additional section to discuss this topic (Line 498-516).

Reviewer 2 Report

Comments and Suggestions for Authors

This manuscript provides a comprehensive review of transcriptome and temporal transcriptome analysis methods in single cells. The authors offer an in-depth discussion of single-cell RNA sequencing (scRNA-seq) advancements and the incorporation of temporal information, which enhances our understanding of dynamic biological processes. While the review is thorough, some areas could benefit from further clarification or additional details. Below are specific comments for improvement.

1. Could the authors provide a summary of bioinformatic tools specifically focused on analyzing temporal single-cell transcriptomes? A dedicated section or table summarizing these key studies would help readers quickly understand the current landscape and advances in temporal analysis within single-cell transcriptomics.

2. In section 3.4, Could the authors consider using a table to illustrate the differences between temporal single-cell transcriptomics and traditional sc-RNAseq in clustering and trajectory analysis?

3. Have there been any benchmark studies comparing the strengths and limitations of different library preparation methods for temporal single-cell transcriptomics? If available, referencing such articles would provide readers with insights into the relative performance of these methods and could guide them in selecting the most suitable approach for their research.

4. Could the authors discuss the minimum and maximum time span that can be measured with temporal single-cell transcriptomics?

Author Response

This manuscript provides a comprehensive review of transcriptome and temporal transcriptome analysis methods in single cells. The authors offer an in-depth discussion of single-cell RNA sequencing (scRNA-seq) advancements and the incorporation of temporal information, which enhances our understanding of dynamic biological processes. While the review is thorough, some areas could benefit from further clarification or additional details. Below are specific comments for improvement.

We thank the reviewer’s comment that our manuscript is thorough and offers an in-depth discussion of scRNA-seq and temporal scRNA-seq. We sincerely appreciate the reviewers’ constructive suggestions to further improve this manuscript. We provide a point-to-point response as below.

  1. Could the authors provide a summary of bioinformatic tools specifically focused on analyzing temporal single-cell transcriptomes? A dedicated section or table summarizing these key studies would help readers quickly understand the current landscape and advances in temporal analysis within single-cell transcriptomics.

We thank the reviewer for the suggestion. Unlike scRNA-seq data analysis pipelines, temporal scRNA-seq data analysis pipelines are not that well-developed, largely consisting of customized scripts deposited at Github. We have now summarized the pipelines in Table 2.

  1. In section 3.4, Could the authors consider using a table to illustrate the differences between temporal single-cell transcriptomics and traditional sc-RNAseq in clustering and trajectory analysis?

We thank the reviewer for the suggestion. We have now provided Table 4, focusing on data matrixes provided by scRNA-seq and temporal scRNA-seq, for clustering and trajectory analysis.

  1. Have there been any benchmark studies comparing the strengths and limitations of different library preparation methods for temporal single-cell transcriptomics? If available, referencing such articles would provide readers with insights into the relative performance of these methods and could guide them in selecting the most suitable approach for their research.

We thank the reviewer for the suggestion. However, there are no benchmarking studies comparing temporal scRNA-seq so far. We also pointed this out in the manuscript (Line 377-378) by stating “While a number of temporal scRNA-seq methods have been developed, a systematic comparison between them is still lacking.

  1. Could the authors discuss the minimum and maximum time span that can be measured with temporal single-cell transcriptomics?

We thank the reviewer for the suggestion. We have now included an additional section in the revised manuscript to discuss the technical concerns and solutions of temporal scRNA-seq (Line 498-516). In this section, we have answered this question by stating “An optimal incorporating rate ranging from 1% to 10% could result from titrations of 4sU concentration (100 µM to 1 mM) and labeling time (15 min to 24h), with consideration of cytotoxicity and RNA half-life, respectively”.

Round 2

Reviewer 1 Report

Comments and Suggestions for Authors

The authors have well addressed most of my concerns. I recommend publication.